# SoundCount: Sound Counting from Raw Audio with Dyadic Decomposition Neural Network

## Abstract

In this paper, we study an underexplored, yet important and challenging problem: counting the number of distinct sound in raw audio characterized by a high degree of polyphonicity. We do so by systematically proposing a novel end-to-end trainable neural network (we call DyDecNet, comprising of a dyadic decomposition front-end and backbone network), and quantifying the difficulty level of counting depending on sound polyphonicity. Unlike existing audio-processing methods that uniformly apply a set of frequency-selective filters on the raw waveform in a one-stage manner to get time-frequency (TF) representation, our dyadic decomposition front-end progressively decomposes the raw waveform dyadically along the frequency axis to obtain TF representation in multi-stage, coarse-to-fine manner. Each intermediate waveform convolved by a parent filter is further processed by a pair of child filters that evenly split the parent filter's carried frequency response, with the higher-half child filter encoding the *detail* and lower-half child filter encoding the *approximation*. We further introduce an energy gain normalization to normalize sound loudness variance and spectrum overlap, and apply it each intermediate parent waveform before feeding it to the two child filters. We argue that such dyadic decomposition front-end better characterizes sound polyphonicity and concurrency that commonly exist in sound counting task, while introducing negligible extra computational cost. To better quantify sound counting difficulty level, we further design three polyphony-aware metrics: *polyphony ratio*, *max polyphony* and *mean polyphony*. We test DyDecNet on three main sound datasets from different domains: bioacoustic sound (both synthetic and real-world sound), telephone-ring sound and music sound. Comprehensive experiment results show our method outperforms existing sound event detection (SED) methods significantly. The dyadic decomposition front-end can be used as a general front-end by existing methods to improve their performance accordingly.

## 1 Introduction

Suppose you went to the seaside and heard a cacophony of seagulls, squawking and squabbling. An interesting question that naturally arises is whether you can tell the number of seagulls flocking around you from the sound you heard? Although a trivial example, this sound "crowd counting" problem has a number of important applications. For example, passive acoustic monitoring (PAM) is widely used to record sounds in natural habitats, which provides measures of ecosystem diversity and density [2, 15, 12]. Sound counting helps to quantify and map sound pollution by counting the number of individual polluting events [4]. It can also be used in music content analysis [24]. Despite its importance, research on sound counting has far lagged behind than its well-established crowd counting counterparts from either images [49, 46], video [29] or joint audio-visual [22].

We conjecture that the lack of exploration stems from three main factors. First, sound counting has long been thought of as an over-solved problem by sound event detection (SED) methods [35, 9, 1, 19], in which SED goes further to identify each sound event's (e.g. a bird call) start time, end time and semantic identity. Sound counting number then becomes easily accessible by simply adding up all detected events. Secondly, current SED only tags whether a class of sound event is present within a window, regardless of the number of concurrent sound sources of the same class like a series of baby crying or multiple bird calls [41]. Thirdly, labelling acoustic data is technically-harder and more time-consuming than labelling images, due to the overlap of concurrent and diverse sources. The lack

of well-labelled sound data in crowded sound scenes naturally hampers research progress. Existing SED sound datasets [1, 20] capture simple acoustic scenarios with low polyphony and where the event variance is small. The simplified acoustic scenario in turn makes sound counting task by SED methods tackleable. But when the sound scene becomes much more complex with highly concurrent sound events, SED methods soon lose their capability in discriminating different sound events [38, 9]. Therefore, a study specific for sound counting problem is desirable and overdue.

In this paper, we study the general sound counting problem under highly polyphonic, cluttered and concurrent situation. Whilst the challenges of image-based crowd counting mainly lie in spatial density, occlusion and view perspective distortion, the sound counting challenges are two-fold. Firstly, acoustic scenes are additive mixtures of sound along both time and frequency axes, making counting overlapping sounds difficult (temporal concurrence and spectrum-overlap). Secondly, there is a large variance in event loudness due to spherical signal attenuation with distance (loudness variance). Tackling these challenges require a more elegant method to process sound raw waveform so as to better localize sound in time-frequency domain.

In this paper, we propose a novel dyadic decomposition neural network to learn a sound density representation capable of estimating cardinality directly from raw sound waveform. Unlike existing sound waveform processing methods that all apply frequency-selective filters on the raw waveform in single stage [19, 10, 48, 18, 14], our network progressively decomposes raw sound waveform in a dyadic manner, where the intermediate waveform convolved by each parent filter is further processed by its two child filters. The two child filters evenly split the parent filter's frequency response, with one child filter encoding the waveform *approximation* (the one with the lower-half frequency response) and the other one encoding the waveform *details* (the one with the higher-half frequency response). To accommodate sound loudness variance, spectrum-overlap and time-concurrence, we further propose an energy gain normalization module to regularize each intermediate parent waveform before feeding it to two child filters for further processing. This hierarchical dyadic decomposition front-end enables the neural network to learn a robust TF representation in multi-stage coarse-to-fine manner, while introducing negligible extra computation cost. By setting each filter's frequency cutoff parameters to be learnable and self-adjustable during optimization in data-driven way, the final learned TF representation can better characterize sound existence in time and frequency domain. Following the front-end, we add a backbone network to continue to learn a time framewise representation. Such representation can be used to derive the final sound count number by either directly regressing the count number, regressing density map (the one we choose) or following SED pipeline.

Apart from the network, we further propose three polyphony-aware metrics to quantify sound counting task difficulty level: polyphony ratio, maximum polyphony and mean polyphony. We will give detailed discussion to show the feasibility of three metrics.

We run experiments on four cross-domain sound datasets: a bird sound set (both real-world and synthetic), a telephone-ring sound set (synthetic), and music sound [24] (real-world). Experimental results show our method (DyDecNet) outperforms exiting SED-based methods significantly on both real-world and synthetic dataset. Replacing existing methods' one-stage sound raw waveform processing front-end with our dyadic decomposition front-end dramatically improves their performance accordingly. Since the real-world datasets contain relatively small polyphony level, we specially synthesize a bird sound dataset that contain much higher sound polyphonic level and spectral overlap. The synthesized sound dataset has two sub-sets: one involves four kinds of bird sound (exhibits heterophony); the other has just one kind of sound (this encapsulates homophonic scenario). Experiment on such synthetic dataset helps to test performance under highly polyphonic situation.

In summary, we make three main contributions: **First**, propose dyadic decomposition front-end to decompose the raw waveform in a multi-stage, coarse-to-fine manner, which better handles loudness variance, spectrum-overlap and time-concurrence. **Second**, propose a new set of polyphony-aware evaluation metrics to comprehensively and objectively quantify sound counting difficulty level. **Third**, Show the efficiency and generalization of DyDecNet on sound datasets across different domains.

## 2 RELATED WORK

Crowd counting from images or audio-visual has been thoroughly studied in recent years [49, 22], the target of which is to estimate the instance number from very crowded scenes (e.g. pedestrian in train station) that cannot be efficiently handled by object detection methods. The methods approaching image crowd counting chronically evolve from the early detection-based [26] to the later regression-

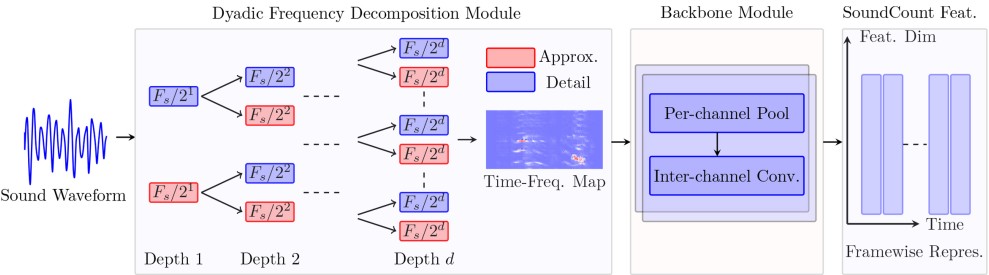

Figure 1: DyDecNet pipeline. We first feed the input raw sound waveform to the dyadic decomposition front-end to learn a time-frequency representation, which is further fed to a backbone neural network to continue to learn framewise representation. Such representation retains time information, so it is general enough to get count number by either regression or SED method. The dyadic decomposition front-end consists of a set of parameterized learnable band-pass filters. Each intermediate waveform processed by a parent filter is further processed by two child filters, with lower-half filter (red color) encoding *approximation* and higher-half filter (light-blue) encoding *details*.

based [11] and density map estimation [27] methods. Accompanying these methods, various neural network architectures have been designed to achieve higher performance.

The counterpart task purely in sound, however, has been nearly ignored. Existing research mainly focus on sound event detection, including spatio-temporal sound event detection (SELD) [19, 18, 1, 10] from a microphone array and temporal sound event detection [7, 39] and high-frequency time series analysis [36]. They often combine convolutional neural networks (CNN) [8] and reccurent neural network [1] to separate sound sources. The datasets they work on are relatively simple, in which the sound scenes are relatively simple and contain few overlapping sound events.

The common way to process raw sound waveform is to first convert the 1D waveform into 2D time-frequency representation so that sound events' frequency property and their variation along time axis are explicitly split out. Most existing methods [10, 1, 7, 39] adopt Fourier transform [14] or Wavelet transform [33] to obtain such 2D representation, in which the whole conversion process is fixed. Some recent work [19, 48, 42, 45, 37] re-parameterize the conversion frequency-selective filters to be learnable so that the whole neural network is able to directly learn from raw sound waveform. Experimental results show enabling the neural network to learn from the raw waveform can often achieve better performance than traditional fixed conversion. These methods, however, convert the raw waveform in a one-stage manner. Our proposed dyadic decomposition neural network instead processes the raw waveform in a dyadic multi-stage manner.

**Dyadic Network** Dyadic representation idea has been initially proposed to represent signal hierarchically [6, 3], in multi-scale manner. Its core idea is to construct a bank of filters (either learnable or fixed) so that different filter extracts different feature at a certain scale or resolution. Summarizing them together leads to more comprehensive and complete analysis. Similar idea has been widely used in computer vision community, including pyramid feature representation for object detection [30] and semantic segmentation [43, 28].

## 3 DYADIC DECOMPOSITION NEURAL NETWORK

Different sound classes typically exhibit different spectral properties. A canonical way to process raw sound waveform is to apply a frequency-selective filter bank $\mathcal{F}_f = \{f_i\}_{i=1}^k$ to project the raw sound waveform onto different frequency bins. Traditional Fourier transform [14] or Wavelet transform [33] construct fixed filter banks in which all filter-construction relevant hyperparameters are empirically chosen and thus may not be optimal for a particular task. Recent methods [19, 48] relax some hyperparameters to be trainable so that the filter bank can be optimized in a data-driven way. A learnable filter bank often leads to better performance than fixed filters. However, all existing methods apply all filters, either learnable or fixed, on the raw waveform in a one-stage manner. Such shallow and one-stage processing may fail to learn powerful and robust representation for sound counting task where large loudness variance and heavy spectrum overlap exist. In our dyadic decomposition framework, we instead adopt a progressive pairwise decomposition strategy to obtain the time-frequency (TF) representation. It learns a TF representation from coarse-grained to fine-grained granularity. Particularly, it consists of a dyadic frontend and a backbone.

## 3.1 DYADIC FREQUENCY DECOMPOSITION FRONTEND

In dyadic frequency decomposition frontend, we construct a set of $D$ hierarchical filter banks $\mathcal{F}_{dyadic}^{D} = \{\mathcal{F}_{2^1}^1, \mathcal{F}_{2^2}^2, \cdots, \mathcal{F}_{2^D}^D\}$. The $d$-th filter bank has $2^d$ filters, each filter is parameterized by a learnable high freuqney-cutoff parameter and a low frequency-cutoff parameter. By cascading these filter banks, we consecutively decompose the raw waveform in frequency domain dyadically, leading to coarse-grained to fine-grained TF representation. Specifically, we denote the dyadic filter banks depth by $D$, in the depth $d$ filter bank $\mathcal{F}_{2^d}^d$, we have $2^d$ filters evenly divide the waveform sampling frequency $F_s$. Therefore, each single filter's frequency response length is $\frac{F_s}{2^d}$, the $i$-th filter $f_i^d$ high frequency cutoff $F_h$ and low frequency cutoff $F_l$ are initialized as,

$$F_h(f_i^d) = \frac{F_s}{2^d} \cdot (i+1), \quad F_l(f_i^d) = \frac{F_s}{2^d} \cdot i \tag{1}$$

From Eqn. (1) we can see that dyadic decomposition frontend forms a complete binary-tree-like structure, in which the filter number doubles and each filter's frequency response length halves as the tree's depth increases by one. The intermediate waveform processed by a "parent" filter is just further processed by its two "childre" filters. The frequency responses of the two children filters evenly split their parent filter's frequency response. The child filter carrying the higher half frequency response encode the parent's processed intermediate waveform's *detail* while the other one carrying the lower half frequency response instead encodes the *approximation*. For example, for the filter $f_i^d$ in the $d$-th filter bank, its frequency response lies in $[\frac{F_s}{2^d} \cdot i, \frac{F_s}{2^d} \cdot (i+1)]$, its two children filters $f_{2i}^{d+1}$ and $f_{2i+1}^{d+1}$ in the depth $d+1$ evenly divide its frequency range, so $f_{2i}^{d+1}$ carries $[\frac{F_s}{2^d} \cdot i, \frac{F_s}{2^d}(i+\frac{1}{2})]$. $f_{2i+1}^{d+1}$ carries $[\frac{F_s}{2^d}(i+\frac{1}{2}), \frac{F_s}{2^d} \cdot (i+1)]$.

With the pre-constructed dyadic decomposition filter banks, we cascade them together to process the raw sound waveform, progressively learning the final TF representation. In our implementation, each filter in dyadic filter banks is a learnable band-pass filter. We adopt rectangular band-pass in frequency domain filter which comprises of a learnable high frequency cutoff parameter $F_h$ and a learnable low frequency cutoff parameter $F_l$. Converting it to time domain through inverse Fourier transform, we get $sinc(\cdot)$ function like filter that is used to convolve with the waveform. For example, the filter $f_i^d$ in Eqn. (1) can be represented as,

$$f_i^d[t, F_h, F_l] = 2F_h sinc(2\pi F_h t) - 2F_l sinc(2\pi F_l t) \tag{2}$$

where $sinc(x) = sin(x)/x$, $t$ indicates the filter's representation at time $t$. $F_h$ and $F_l$ are initialized according to Eqn. (1), but they can be further adjusted during training process. $sinc(\cdot)$ filters have been successfully used in speech recognition [42] and sound event detection and localization [19]. In our dyadic decomposition frontend, each filter from different depth has separate and independent learnable parameters (high frequency cutoff and low frequency cutoff). Moreover, our constructed filter is much longer (1025 in our case) than traditional 1D/2D Conv filters (3 or 5). Its wide length characteristic enables the filter to have wide field-of-view on the raw waveform. Cascading them together allows the filters in later layers (larger depth) to have even wider field-of-view on the input raw waveform. With this advantage, we do not have to model sound event temporal dependency explicitly with RNN network. As a result, the whole dyadic frequency decomposition frontend is fully convolutional and parametrically learnable, it is parameter-frugal and computationally efficient. In practice, the dyadic decomposition frontend depth is 8, so the output TF representation has 256 frequency bins. At the same time, we downsample the intermediate waveform by 2 before feeding it to its two children filters in the initial 5 dyadic filter banks to reduce the memory cost.

## 3.2 ENERGY GAIN NORMALIZATION

We further design an energy gain normalization module to regularize each intermediate waveform before feeding them to the next dyadic filter bank. The motivation of introducing energy gain normalization is two-fold: first, to reduce sound event loudness variance led by sound events' different spatial locations; Second, to reinforce the frontend to learn to better tackle spectrum overlap challenge led by intra-class sound events in the sound scene. Specifically, for the intermediate waveform $W_{f_i^d}$ processed by a dyadic filter $f_i^d$, we first smooth it with a learnable 1D Gaussian kernel $g_i^d$ parameterized by learnable width $\sigma$ to get the corresponding smoothed waveform $W_{g_i^d}$ which just

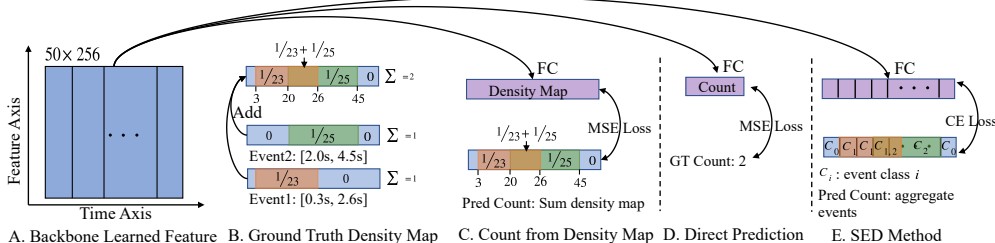

Figure 2: Three counting methods illustration. For density map (sub-fig. C), the sum (or integral) of the density map equals to the count number. We can also direct regress the final count number (sub-fig. D), or use SED method (sub-fig. E). Detailed illustration is in Appendix Fig. V.

contains loudness. We then introduce a learnable automatic gain control parameter $\alpha$ to mitigate sound loudness impact. Furthermore, another two learnable compression parameters $\delta$ and $\gamma$ are introduced to further compress $W_{f_i^d}$. The overall energy gain normalization can be represented as,

$$W_{f_i^d} = \left(\frac{W_{f_i^d}}{(W_{g_i^d})^\alpha} + \delta\right)^\gamma - \delta^\gamma \tag{3}$$

where $\alpha$, $\delta$ and $\gamma$ are learnable parameters. As a result, the energy gain normalization *eg*-Norm is fully learnable and parametersized by four learnable parameters *eg*-Norm$(\sigma, \alpha, \delta, \gamma)$. Practically, each filter in dyadic filter banks is associated with an independent *eg*-Norm module. Similar energy normalization has been successfully used in tasks like keyword spotting [47, 31]. The difference lies in the fact that they apply exponential moving average operation to get smoothed waveform representation, so the computation is very slow because it iterates along the time axis to compute the averaged value step by step. Our proposed energy gain normalized strategy instead adopts a Gaussian kernel to get the smoothed waveform, in which it can be easily implemented as 1D convolution. The dyadic filter visualization and energy normalization module is shown in Fig. 3.

### 3.3 BACKBONE NEURAL NETWORK

We add a lightweight backbone neural network to the frontend neural network to further learn a representation useful for call counting . The backbone network consists of two parts: per-channel pooling and inter-channel 1D convolution. Unlike existing methods [9, 1] that first convert 1D sound waveform into 2D map with fixed FFT-like transform, then learns from the 2D map with 2D Conv. operations, our method directly learn from sound raw waveform with learnable 1D Conv.. Specifically, we downsample each channel separately by assigning each channel with an independent frequency-sensitive learnable filter. We call such learnable downsampling per-channel pooling. It helps to learn sound event's frequency variance along time axis individually. Moreover, we add normal 1D Conv. to achieve inter-channel communication, which enhances the neural network to learn concurrent sound events interaction. Detailed illustration is given in Appendix Table IV. The backbone serves as as the backend to learn framewise representation for counting.

### 3.4 DENSITY MAP AND LOSS FUNCTION

The backbone network discussed above learns a framewise representation $[T_b, F_b]$, where $T_b$ indicates the time steps and $F_b$ indicates feature size. There are three potential ways to derive final sound count number fromthe learned representation: 1. directly regress the count number; 2.SED method: detect sound events first and then aggregate results to get final count; 3. predict the density map. For a sound event with time location $[t_1, t_2]$, its density map is a 1D vector with value $\frac{1}{t_2-t_1}$ during its occurrence time, otherwise is 0. So the count number equals to the vector integral. We show regressing density map produces the best result (see Table 6). We thus adopt the mean squared error (**MSE**) loss during training to directly regress the density map. The comparison of three methods is shown in Fig. 2.

## 4 EVALUATION METRIC DISCUSSION

Mean absolute error (**MAE**) and mean squared error (**MSE**) are two widely used metrics in crowd counting [32, 49]. Specifically, denote the ground truth count and predicted count by $y_i$ and $\hat{y}_i$ respectively, for the $i$-th sound clip. MAE is defined as $\mathrm{MAE} = \frac{1}{N} \sum_{i=1}^{N} |y_i - \hat{y}_i|$, MSE is defined

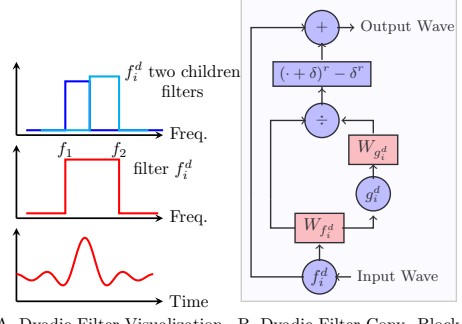
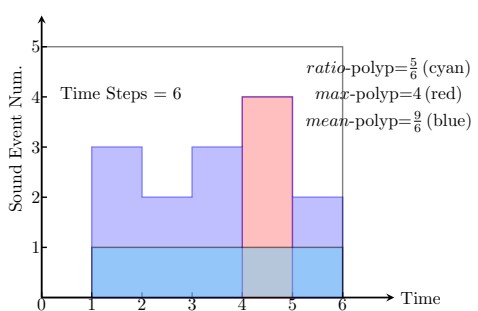

A. Dyadic Filter Visualization    B. Dyadic Filter Conv. Block

Figure 3: Dyadic filter illustration. Left: In time domain, dyadic filter is a *sinc* function curve. In frequency domain, dyadic filter is a rectangular band-pass filter with learnable high frequency $f_2$ and low frequency cutoff $f_1$. The filter's two child filters (left-top) evenly splits the parent filter's frequency response. Right: dyadic filter convolution block. The input waveform is fed to an energy normalization module. Then a skip-connection is added.

Figure 4: The max polyphony level in this sound clip is 4 (show in red, time step 5), so $max$-polyp=4. The $mean$-polyp indicates the purple area, averaging them over time gets $\frac{9}{6}$. $ratio$-polyp measures polyphony (no fewer than two concurrent sound events) existence ratio along the time axis (cyan), so it is $\frac{5}{6}$.

as MSE $= \sqrt{\frac{1}{N}\sum_{i=1}^{N}(y_i - \hat{y}_i)^2}$. We also involve accuracy rate (**AccuRate**) to show the ratio of accurately predicted count. We introduce a tolerance term $p$, where $p = 0$ means predicted count number has to be exactly the same with ground truth number in order to be treated as an accurate counting; $p = 1$ relaxes the constraint so there can be one count mismatch for an accurate counting.

### 4.1 POLYPHONY-AWARE DIFFICULTY QUANTIFICATION

The aforementioned three general metrics do not reflect the impact of sound scene nature on algorithms. We introduce three polyphony-aware metrics to quantify sound counting difficulty level reflected by sound scene nature. The three metrics are time-window invariant so they can be used as general metrics to quantify difficulty level of sound scene of various lengths.

**Polyphony Ratio** ($ratio$-polyp) describes the ratio of polyphony (at least two sound events happen at the same time) over a period of time. It binarizes each time step as either polyphonic or non-polyphonic (monophoinc or silent) so the value lies between $[0, 1]$.

**Maximum Polyphony** ($max$-polyp) focuses on the maximum polyphony level over a time period. It is motivated by the fact that human's capability in discriminating different sound events reduces seriously when the number of temporal-overlapping sound event number increases. It is a positive integer and helps us to understand an algorithm's capability in tackling polyphony peak.

**Mean Polyphony** ($mean$-polyp) instead focuses on the averaging level of polyphony involved within a time period. It is designed to reflect algorithm's capability in tackling the average polyphony level over an arbitrary time window.

Given $T_n$ time steps sound vector $[p_1, p_2, \cdots, p_{T_n}]$, where $p_i \geq 0$ is the sound event number happening at time step $T_i$. The three metrics are defined as,

$$ratio-\text{polyp} = \frac{\sum_{i=1}^{n} \mathbb{1}_2(p_i)}{n}; \ max-\text{polyp} = \max_{i=1,\cdots,n} p_i; \ mean-\text{polyp} = \frac{\sum_{i=1}^{n} \max(p_i - 1, 0)}{T_n}$$
(4)

where $\mathbb{1}_2(p_i)$ is an indicator function, it is 1 if $p_i \geq 2$, otherwise 0. With the three quantifying metrics, we can report the general metrics (MAE, MSE) against various difficulty levels.

## 5 EXPERIMENT

We run experiments on six sound datasets derive from four commonly seen domains.

1. **Bioacoustic Sound**. We focus on bird sound as bird sound is ubiquitous is most terrestrial environment with distinctive vocal acoustic properties. Specifically, we test three datasets: one real-world NorthEastUS_Bird [12] dataset and other two synthesized datasets: Polyphony4Birds (for heterophony test) and Polyphony1Bird (for homophony test). NorthEastUS_Bird data is recorded in nature reserve in northeastern of United States. It encompasses 385 minutes of dawn chorus recordings collected in July 2018, with a total of 48 bird species. The average bird sound temporal length is very short (less than 1s) and the polyphony level ($max$-polyp and $mean$-polyp) is small.

   To test performance under highly polyphonic situation, we synthesize two bird sound datasets. Specifically, The first dataset contains four sounds: junco, American redhead, eagle, and rooster from copyright-free website `findsounds.com`. We call it Polyphony4Birds (heterophony test). The second dataset contains one sound: rooster. We call it Polyphony1Bird (homophony test).

2. **Indoor Sound**. We count telephone ring sound, the telephone ring seed sound comes from the same copyright-free website. We follow Polyphony1Bird synthesis procedure except synthesize in a much smaller room ($10m \times 10m \times 3m$) to reflect room reverberation effect.

3. **Outdoor Sound**. We count car engine sound, as it is widely heard in outdoor urban scenario. The car engine seed sound also comes from the same copyright-free website. We also follow Polyphony1Bird synthesis procedure to create car engine dataset.

4. **Music Sound**. We use OpenMic2018 dataset [24]. The target is to count the musical instrument class number being played in the audio clip, regardless of the number of times each single musical instrument class being played in the audio clip. This dataset contains 20 musical instrument categories, but we do not know each instrument's playing start time and end time, but instead the total sound count number within each clip. Therefore, we directly regress the number.

Table 1: The comparison on the six datasets, in terms of data size, sound event class number and polyphony level.

| Data | Size | Class | mean-poly | max-poly |
|---|---|---|---|---|
| NorthEastUS [12] | 6.41 h | 48 | 0.1 | 4 |
| Polyphony4Birds | **55.56 h** | 4 | **1.244** | **9** |
| Polyphony1Bird | **55.56 h** | 1 | **1.975** | **9** |
| Telephone Ring | 55.6 h | 1 | **1.975** | **9** |
| Car Engine | 55.6 h | 1 | **1.975** | **9** |
| OpenMic [24] | 55.6 h | 20 | n/a | n/a |

The direct comparison between the six datasets is given in Table 1. We highly refer to Appendix Sec. B for more discussion about the data synthesis process.

**Comparing Methods**: We compare our framework with two main method categories: 1) traditional deterministic signal processing methods, including Librosa-onset and Aubio-onset; 2) SED-based Methods. **Librosa-onset** [34] provides an onset/offset detection method for music note detection. It measures the uplift or shift of spectral energy to decide the starting time of a note. We use its onset/offset detection ability to count sound event number. **Aubio-onset** [5] achieves pitch tracking by aligning period and phase of the Mel spectrogram. We use its pitch tracking to count.

SED-based methods build on traditional fixed TF representation, such as short time Fourier transform (STFT) and LogMel. The TF representation is treated as a 2D image to be processed by a sequence of 2D Conv. operators. GRU [13] and LSTM [21] are often adopted to model temporal dependency. we compare three typical SED methods: 1) **CRNNNet** [9] consists of 2D Conv. to learn multiple compressed TF representations from the input TF map. Then it concatenates them together along the frequency dimension and further feed it to LSTM [21] to learn framewise representation. 2) **DND-SED** [16] instead adopts depthwise 2D convolution and dilated convolution to avoid using RNN. 3) **SELDNet** [1] is originally used for joint sound event detection and localization. It adopts 2D Conv. to convolve the 2D TF map, and bidirectional GRU to model temporal dependency. The three comparing methods' network architectures are slightly adjusted to fit our dataset. We call our method **Dy**adic **Dec**omposition **Net**work (DyDecNet).

**Implementation Detail and Experiment Configuration** For all the six datasets, all input audios are segmented into 5 second long clips, with sampling rate 24 k Hz. So the input waveform has 120,000 data points and is normalized into $[-1, 1]$. We train the models with Pytorch [40] on TITAN RTX

Table 2: MSE and MAE results on the six datasets. We leave the Accuracy Rate metric in the Appendix due to space limitation.

| Dataset | OpenMIC | | NorthEastUS | | Polyphony4Birds | | Polyphony1Bird | | TelephoneRing | | CarEngine | |
|---|---|---|---|---|---|---|---|---|---|---|---|---|
| Method | MSE↓ | MAE↓ | MSE↓ | MAE↓ | MSE↓ | MAE↓ | MSE↓ | MAE↓ | MSE↓ | MAE↓ | MSE↓ | MAE↓ |
| Librosa-onset | 25.3 | 4.00 | 2.31 | 1.65 | 28.3 | 4.09 | 37.63 | 5.5 | 30.03 | 4.50 | 33.13 | 4.51 |
| Aubio-onset | 7.40 | 1.72 | 4.91 | 1.74 | 8.43 | 1.91 | 35.33 | 5.27 | 33.20 | 4.22 | 35.13 | 4.76 |
| SELDNet [17] | 0.90 | 1.37 | 1.35 | 1.79 | 0.92 | 1.41 | 0.89 | 1.19 | 0.97 | 1.30 | 0.92 | 1.23 |
| CRNNNet [9] | 0.71 | 1.00 | 1.33 | 1.77 | 0.74 | 1.10 | 0.87 | 1.16 | 0.92 | 1.31 | 0.86 | 1.15 |
| DND-SED [16] | 0.93 | 1.27 | 1.19 | 1.64 | 0.95 | 1.34 | 1.04 | 1.27 | 1.23 | 1.34 | 1.00 | 1.21 |
| DyDecNet | **0.32** | **0.72** | **0.85** | **1.19** | **0.46** | **0.92** | **0.54** | **0.85** | **0.58** | **0.89** | **0.54** | **0.87** |

GPU. Network architecture of DyDecNet is shown in Appendix Table IV. To train the neural network, we adopt Adam optimizer [25] with an initial learning rate 0.001 which decays every 20 epochs with a decaying rate 0.5. Overall, we train 60 epochs. We train each method 10 times independently and report the mean value and standard deviation. We do not report the standard deviation explicitly in the table because we find them very small (about 0.03). We first train the comparing SED methods with both their suggested training strategy and our training strategy, then choose the one with the better performance as the final result. For the energy gain normalization we initialize them as $\alpha = 0.96$, $\delta = 2.$, $\gamma = 0.5$, $\sigma = 0.5$. The batchsize is 128.

## 5.1 EXPERIMENTAL RESULT

The quantitative result on MSE MAE and is is given in Table 2, and the accuracy rate result is given in Table II in Appendix Material. From the two tables we can learn that our proposed DyDecNet outperforms both classic signal processing deterministic methods and existing SED methods by a large margin. Our framework is better than the baselines discussed in this paper in both real-world and synthesized sound scenes. It is capable of learning powerful representation from both weak sound signals (NorthEastUS_Bird dataset), highly polyphonic (Our synthesized four datasets) and heavy spectrum-overlapping, loudness-varying sound events.

At the same time, we also observe that the two signal processing deterministic methods (Librosa-onset and Aubio-onset) generate the worst result over both SED based methods and DyDecNet. The higher of the polyphony level of the dataset, the worse performance the two deterministic methods lead to. For example, in NorthEastUS_Bird dataset with a relatively smaller polyphony level, Librosa-onset and Aubio-onset generate relatively good performance with accuracy rate ($p = 1$) reaching 0.58. In our synthesized two datasets with much higher polyphony levels, however, their accuracy drops significantly to near zeros. It thus shows traditional signal processing methods do not fit for sound event counting from crowded acoustic scenes.

Among the three datasets, SED-based methods and DyDecNet produce decreasing performance on Polyphony4Birds, Polyphony1Bird and NorthEastUS_Birds dataset, respectively. The largest performance drop is observed on real-world NorthEastUS_Birds dataset, which shows counting from real-world dataset is a tough task that desires more future attention. Spectrum-overlap led by intra-class sound events is another potential challenge (better performance on Polyphony4Birds than Polyphony1Bird) that may need more work to tackle it.

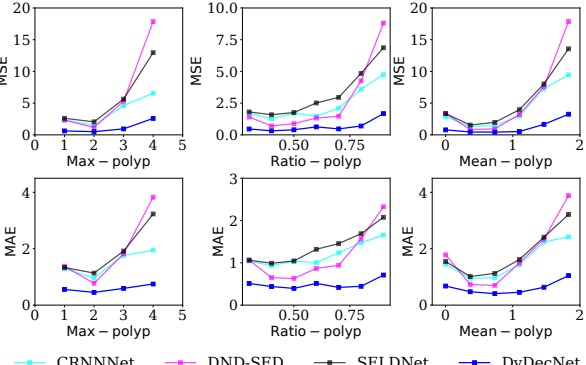

Figure 5: MSE and MAE variation against $max$-polyp, $ratio$-polyp and $mean$-polyp on NorthEastUS_Bird dataset. More results are in Appendix.

The MSE and MAE variation against $max$-polyp, $ratio$-polyp and $mean$-polyp difficulty level on NorthEastUS_Bird are shown in Fig. 5. We can observe that our proposed three metrics $max$-polyp, $ratio$-polyp and $mean$-polyp are effective ways to accurately quantify sound counting tasks difficulty level. The three metrics has observed dramatic performance drop as their the difficulty

Table 3: Ablation study on dyadic decomposition efficiency discussion: we compare existing methods with and without dyadic decomposition frontend.

| Method | MSE↓ | MAE↓ |
|---|---|---|
| SELDNet [17] | 1.35 | 1.79 |
| SELDNet_Dydec | **1.05** | **1.43** |
| CRNNNet [9] | 1.33 | 1.77 |
| CRNNNet_Dydec | **1.20** | **1.51** |
| DND-SED [16] | 1.19 | 1.64 |
| DND-SED_Dydec | **0.89** | **1.40** |

Table 4: Ablation study on traditional T-F feature for counting task: DyDecNet's dyadic decompostion frontend is replaced by various classic T-F features extractors, such STFT, LogMel, MFCC and Gabor.

| Method | MSE↓ | MAE↓ |
|---|---|---|
| DyDecNet_STFT | 1.35 | 1.51 |
| DyDecNet_LogMel | 1.33 | 1.50 |
| DyDecNet_MFCC | 1.32 | 1.49 |
| DyDecNet_Gabor | 1.33 | 1.48 |
| DyDecNet | **0.85** | **1.19** |

Table 5: Ablation study on various DyDecNet variants.

| Method | MSE↓ | MAE↓ |
|---|---|---|
| DyDecNet_SingScale | 1.22 | 1.43 |
| DyDecNet_BN | 1.07 | 1.25 |
| DyDecNet_noNorm | 1.15 | 1.37 |
| DyDecNet | **0.85** | **1.19** |

Table 6: Ablation study on various counting method.

| Method | MSE↓ | MAE↓ |
|---|---|---|
| DyDecNet_RegCount | 1.03 | 1.39 |
| DyDecNet_SED | 2.09 | 3.06 |
| DyDecNet | **0.85** | **1.19** |

level increases. Nevertheless, DyDecNet remains as the best one among all the three difficulty levels, showing DyDecNet outperforms the comparing methods under difficult levels discussed in this paper.

## 5.2 ABLATION STUDY

We do ablation study on NorthEastUS_Bird data.

**First**, disentangling our proposed framework's dyadic decomposition frontend and backbone network so as to figure out their individual contribution. To this end, on the one hand, we concatenate dyadic decomposition frontend to the three SED methods backbone networks so that they can learn TF representation from raw waveform. We call them SELDNet_dydec, CRNNNet_dydec and DND-SED_dydec respectively. On the other hand, we feed our backbone neural network with fixed pre-extracted TF features, including short time Fourier transform (STFT), LogMel, MFCC and Gabor Wavelet filter. We call them DyDecNet_STFT, DyDecNet_LogMel and DyDecNet_MFCC, DyDecNet_Gabor, respectively. The results are in Table 3 and 4. We can observe that: 1) replacing traditional fixed TF feature with dyadic decomposition frontend significantly improves the performance (Table 3). The gain stems from two-fold: our dyadic decomposition frontend enables the network to directly learn from the raw waveform so that all frequency-selective filters are adjustable during training process. Second, the dyadic progressive decomposition enables the neural network to learn robust representation for sound counting. Similarly, a huge performance drop is observed if we let our proposed backbone neural network to learn from traditional fixed TF features (Table 4). Therefore, it shows that both the dyadic decomposition frontend and backbone neural networks are important for sound counting task.

**Second**, we want to figure out if the dyadic decomposition is essential for sound counting, and the importance of energy normalization block. We test three variants: our network with simply single scale decomposition which means applying all filters on the raw waveform (DyDecNet_SingScale) which helps validate necessity of hierarchical dyadically decomposition framework; replacing Energy-normalization module with traditional batch normalization [23] (DyDecNet_BN); without any normalization (DyDecNet_noNorm). The result is in Table 5, from which we can clearly observe that either removing energy normalization or replacing it with batch normalization significantly reduces the performance. It thus shows the importance of energy normalization.

**Lastly** To show the effectiveness of density map, we run two ablation studies to directly regress the final count number or to firstly detect the sound events. From the result in Table 6, we can conclude that directly regressing sound event count number leads to inferior performance than estimating density map. Treating it as a sound event detection problem leads to the worst performance.

Another ablation study on the impact of energy gain normalization on traditional TF feature is presented in Appendix Sec. D.5. We refer reader to this section for more details.

**Limitation Discussion and Conclusion** We do not discuss using microphone-array for enhanced counting, nor test our dyadic decomposition front-end for other acoustic tasks (e.g. source separation). Another limitation is that we just used one instance for each sound category in our synthetic dataset, which does not reflect the real scenario. A more convincing dataset is to involve as many diverse instances for each sound as possible, it also remains as future work.

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

APPENDIX

## A  SOUND COUNTING PROBLEM DEFINITION

Given a mono-channel $T$ seconds raw sound waveform $x(t)$ sampled at a fixed sampling rate $F_s$, the sound recording has recorded $N$ independent sound events $E = \{E_i = (t_s, t_e)\}_{i=1}^N$, each single sound event freely undergoes either stationary or moving motion in the open area. The target is to design a neural network $\mathcal{N}$ parameterized by $\theta$ to predict sound event number $N$ from raw sound waveform $N = \mathcal{N}(\theta|x(t))$. In our formulation, the counting process is class-agnostic, so all sound events are treated as instances to count, regardless of their classes.

Three challenges make it a challenging task: 1) **Large Datasize**: microphone usually records sound at a high frequency rate (*i.e.* 24 kHz), resulting in large data size in the raw waveform. It thus requires more accessible filters with few parameters and computation cost to process the raw sound waveform. 2) **Concurrent Sound Events (polyphony)**: sound events freely overlap both spatially and temporally, resulting highly polyphonic sound recording. It is a tough task to separate them apart from compressed 1D waveform. 3) **Loudness Variance and Spectrum Overlap**: sound events of the same class but different spatial location have large variance in their received loudness. They also have heavy spectrum overlap in the frequency domain. The above issues make counting a tough task.

## B  MORE DISCUSSION ON DATASET CREATION

### B.1  MOTIVATION OF POLYPHONY4BIRDS AND POLYPHONY1BIRD DATASET CREATION

Our motivation of synthesizing Polyphony4Birds and Polyphony1Birds are three-fold:

1. NorthEastUS_Bird dataset has as many as 48 different kind of bird categories. It helps to test various methods' capability in tackling high bird diversity challenge.

2. Polyphony4Birds dataset contains 4 kinds of bird sounds, but in much higher polyphony level (in terms of $ratio$-polyp, $max-$polyp and $mean$-polyp). It helps us to test various methods' capability in tackling limited bird categories but high polyphony level (heterophony test).

3. Polyphony1Bird dataset contains 1 bird sound class in much higher polyphony level. This dataset involves heavy spectrum-overlap (due to the temporal inter-category bird sounds overlap), so it helps to test various methods' capability in tackling high spectrum-overlap and high-polyphony challenge (homophony test).

In Polyphony4Birds dataset, 4 is an arbitrary number. We experimentally find involving 4 bird sounds is representative enough for heterophony test. We note that there are some other relevant public bird sound dataset [44, 20, 9], but we find they are not suitable for our study. For example, in TUT-SED 2009 data [20], the polyphony-level is small and the involved bird sound usually lasts too long (not temporally separable and countable). Similarly, the Bird Audio Detection challenge (BAD challenge) [44] contains highly-sparse bird chirps (very small polyphony-level sound). Moreover, the two real-world bird sound datasets [20, 44] do not provide bird sound start time and end time label, so they are suitable for our study. The other synthesized dataset TUT-SED Synthetic 2016 [9] also contains very limited samples of high polyphony. The direct comparison between these datasets is given in Table I, from which we can see our created two datasets enjoy much higher polyphony-level, making them more suitable for our sound counting task.

Table I: Comparison between various sound dataset, where "n/a" means not available.

| Dataset | Data Source | Size | Event Classes | $mean$-polyp | $max$-polyp |
|---|---|---|---|---|---|
| BAD Challenge [44] | Natural | 23 h | 2 | n/a | 3 |
| TUT-SED 2009 [20] | Natural | 18.9 h | 61 | n/a | 6 |
| TUT-SED Synthetic 2016 [9] | Synthetic | 9.3 h | 16 | 0.659 | 5 |
| NorthEastUS_Bird [12] | Natural | 6.41 h | 48 | 0.1 | 4 |
| Ours Polyphony4Birds | Synthetic | **55.56 h** | 4 | **1.244** | **9** |
| Ours Polyphony1Bird | Synthetic | **55.56 h** | 1 | **1.975** | **9** |

Table II: Accuracy Rate results on the six datasets.

| Dataset / Method | OpenMIC | | NorthEastUS | | Polyphony4Birds | | Polyphony1Bird | | TelephoneRing | | CarEngine | |
|---|---|---|---|---|---|---|---|---|---|---|---|---|
| | Accu. Rate | | Accu. Rate | | Accu. Rate | | Accu. Rate | | Accu. Rate | | Accu. Rate | |
| | p = 0 | p = 1 | p = 0 | p = 1 | p = 0 | p = 1 | p = 0 | p = 1 | p = 0 | p = 1 | p = 0 | p = 1 |
| Librosa-onset | 0.05 | 0.03 | 0.20 | 0.58 | 0.01 | 0.02 | 0.01 | 0.02 | 0.01 | 0.01 | 0.01 | 0.01 |
| Aubio-onset | 0.09 | 0.22 | 0.12 | 0.32 | 0.08 | 0.21 | 0.01 | 0.09 | 0.01 | 0.02 | 0.01 | 0.08 |
| SELDNet [17] | 0.49 | 0.75 | 0.25 | 0.62 | 0.48 | 0.71 | 0.33 | 0.81 | 0.28 | 0.69 | 0.31 | 0.78 |
| CRNNNet [9] | 0.49 | 0.85 | 0.26 | 0.64 | 0.47 | 0.82 | 0.35 | 0.81 | 0.27 | 0.76 | 0.30 | 0.80 |
| DND-SED [16] | 0.47 | 0.70 | 0.28 | 0.67 | 0.43 | 0.68 | 0.24 | 0.71 | 0.18 | 0.63 | 0.20 | 0.68 |
| DyDecNet | **0.68** | **0.92** | **0.40** | **0.82** | **0.70** | **0.88** | **0.55** | **0.92** | **0.53** | **0.89** | **0.57** | **0.94** |

Table III: Comparison of Various Methods. The network block column labels are: 1. 1D Conv, 2. 2D Conv, 3. GRU, 4. LSTM, 5. Depthwise Conv, 6. Dilated Conv. 7. FC, 8. Bi-LSTM, 9. Bi-GRU. The inference time in tested on Intel(R) Core(TM) i9-7920X CPU, we report the average time of 100 independent tests with one 5s waveform.

| Method Name | Input | Param Size | Network Block | Inf. Time | end2end trainable? |
|---|---|---|---|---|---|
| Librosa-onset | Raw Waveform | - | - | 0.1s | ✗ |
| Aubio-onset | Raw Waveform | - | - | 0.1s | ✗ |
| DND-SED [16] | STFT/LogMel | 6.9 M | 2, 5, 6, 7 | 3.0 s | ✗ |
| CRNNNet [9] | STFT | 4.1 M | 2, 3, 8 | 3.3 s | ✗ |
| SELDNet [17] | LogMel | 0.8 M | 2, 3, 7, 9 | 1.2s | ✗ |
| DyDecNet | Raw Waveform | 3.9 M | 1, 7 | 2.7 s | ✔ |

### B.2 How to Simulate Open Area Environment

We collect 4 seed sounds from copyright-free website [1]: junco, American redhead, eagle, and rooster. To maximally reflect outdoor scenario, we simulate a large openarea environment $[100m, 100m, 100m]$ with one microphone at $[50m, 50m, 1m]$. The wall is associated with high sound absorption coefficient, so the reverberation is negligible so as to resemble outdoor open area scenario. We introduce a random SNR (Signal-to-Noise Ratio) at two Gaussian means ($-33$ decibels and $-20$ decibels) at the microphone receiver. We put each seed sound at a random 3D spatial location and a random start time to imitate natural bird sounds that emit sound from a random location and random start time. A post-processing step is added to keep dataset balance between various polyphony-level metrics.

## C More Discussion on Comparing Methods

More detailed comparison between various methods is given in table III. We can see that our proposed DyDecNet is lightweight and directly learns from sound raw waveform (so it is end-to-end trainable). It thus strikes a good balance between model performance and model efficiency (inference time).

## D More Experiment Result Discussion

### D.1 Experiment on NorthEastUS_Bird Dataset and Telephone Ring Dataset

More detailed experimental result (MAE variation) on NorthEastUS_Bird is given in Fig. 5, from which we can observe that with the increasing of $max$-polyp, $ratio$-polyp and $mean$-polyp, all methods (including our DyDecNet) reduces their performances. The three comparing methods (CRNNNet [9],DND-SED [16], and SELDNet [17]) have observed sharp performance drop when the our proposed three sound counting difficulty levels increases, whereas our proposed DyDecNet largely mitigates the challenge caused by higher counting difficulty level (the blue line increases slightly as the counting difficulty level increases). It thus shows 1) our proposed $max$-polyp, $ratio$-polyp and $mean$-polyp are capable of accurately measuring sound counting task difficulty level from different perspectives; 2) our proposed DyDecNet is capable of mitigating these sound counting difficulties.

---

[1]see https://www.findsounds.com/

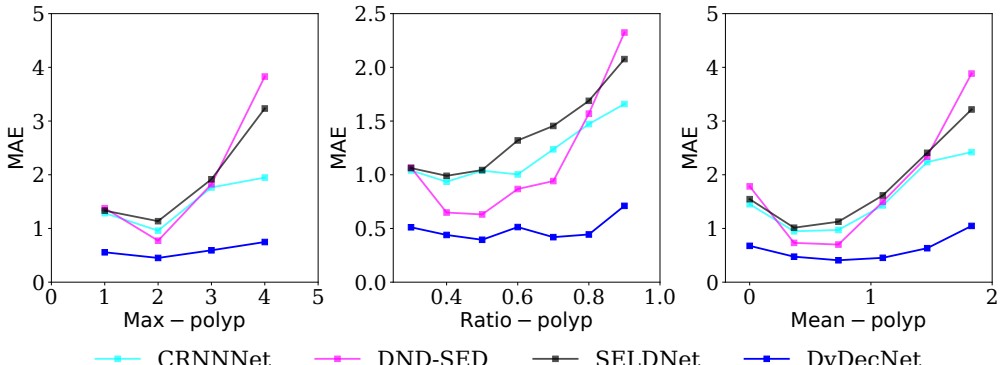

Figure I: MAE variation against $max$-polyp, $ratio$-polyp and $mean$-polyp on NorthEastUS_Bird Dataset.

## D.2 MORE RESULT ON POLYPHONY1BIRD AND POLYPHONY4BIRDS DATASETS

We also provide the detailed results for Polyphony1Bird and Polyphony4Birds in Fig. II and Fig. III, respectively. They contain the accuracy rate, MSE and MAE variation against $max$-polyp, $ratio$-polyp and $mean$-polyp. From the two figures, we can get similar conclusion as of NorthEastUS_Bird dataset (Fig. I): with the increasing of $max$-polyp, $ratio$-polyp and $mean$-polyp, all methods' performance gradually reduces. Our proposed DyDecNet stays as the best-performing one under all sound counting difficulty level metrics. Specifically, we can see that:

- All methods give the best performance on Polyphony4Birds dataset, second best performance on Polyphony1Bird dataset, and the worst performance on NorthEastUS_Bird dataset. It thus shows 1) spectrum-overlap due to high inter-class sound overlap temporally (represented by Polyphony1Bird dataset) remains as a challenge for sound counting task. 2) sound counting in open area where noise pollution, high sound diversity (in our case, diversity means bird categories, we have 48 bird classes in NorthEastUS_Bird dataset), and small labelled data availability exist remains as another challenge for sound counting task. We hope to attract more researchers to consider sound counting task in more challenging scenario.

- We do not observe such sharp performance drop (as we observed on NorthEastUS_Bird dataset) on our two synthetic datasets, which is in contrast with the real-world dataset NorthEastUS_Bird. It thus shows real-world sound counting task becomes increasingly challenging when our proposed three sound counting difficulty level metrics increase. We guess the large model and large training dataset are needed to achieve better performance, which can be treated as a future research direction.

## D.3 COUNTING ON MORE BIRDS CLASSES

In the main paper, our two synthetic datasets Polyphony4Birds and Polyphony1Bird have just involved limited bird classes (up to 4). We naturally want to figure out the performance of all methods (including DyDecNet and the other three comparing methods) under more bird classes situation. We thus follow more the same data creation procedure to synthesize four extra datasets. They contain 2/6/8/10 bird classes, respectively. The extra bird seed sound classes are collected from `findsound.com` too. The quantitative result is given in table V, from which we can learn that all comparing methods have observed performance increasing when the bird classes reach to 6 (values in bold font), then performance decline when bird classes increase to 8 or 10; DyDecNet reaches the best performance around 8 bird classes, then begin to decline. It thus shows: 1) all methods can successfully handle a reasonable amount of bird classes (in our case, maximum bird classes are 8), given the model parameter size budget (less than 10 M) discussed in this paper. When we have to handle much larger bird diverse classes, we might need much larger model (which remains as a future research topic to figure out the relationship between model size and sound counting class diversity); 2) Our proposed DyDecNet exhibits strong capability sound counting in diverse bird classes than the three comparing methods (it reaches the best performance at a higher bird classes (8 bird class)).

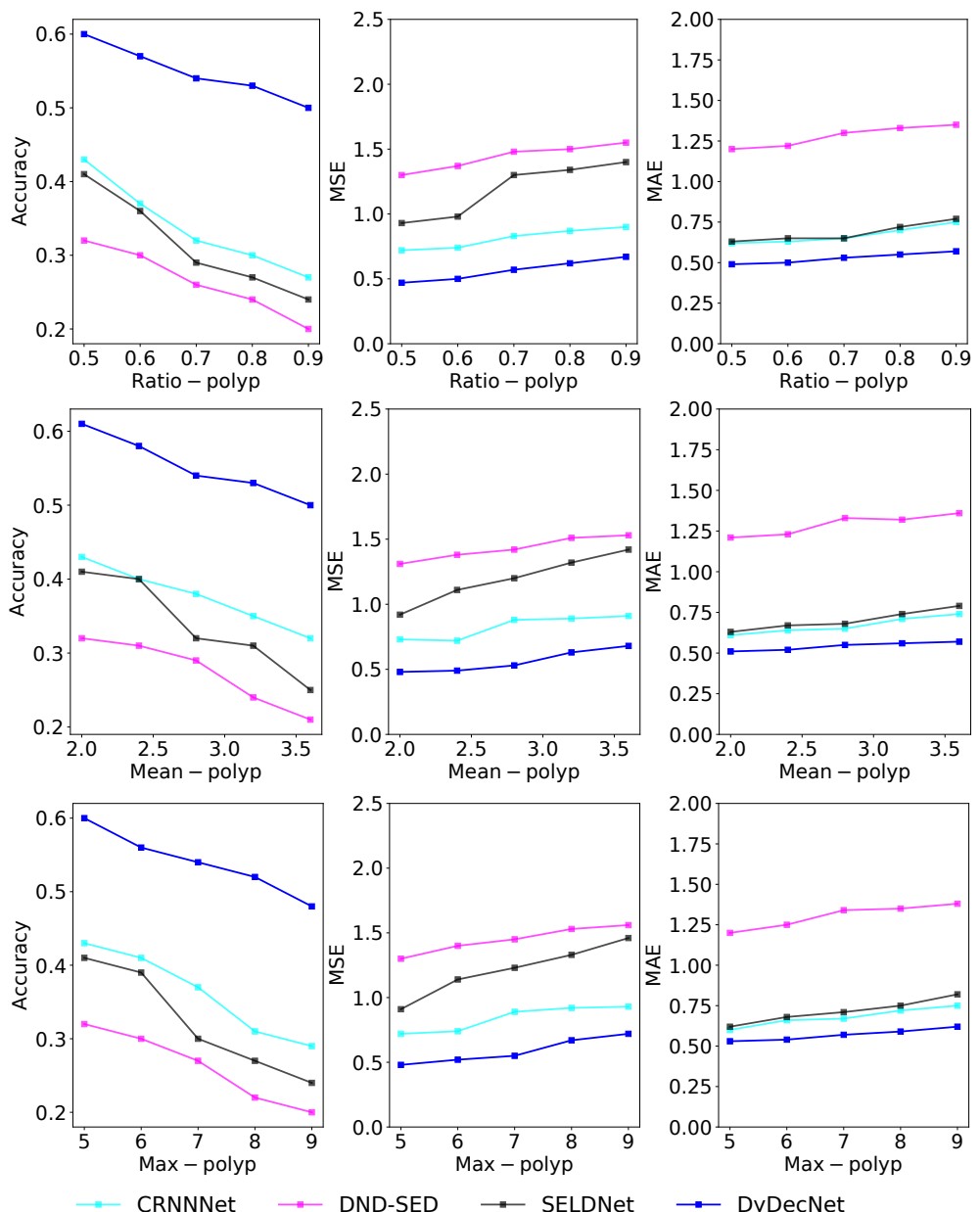

Figure II: AccuRate/MSE/MAE variation against $max$-polyp, $ratio$-polyp and $mean$-polyp on Polyphony1Bird Dataset.

## D.4 NOISE DISCUSSION

In order to show DyDecNet and other comparing methods' performance under different noise interference level, we show the performance on Polyphony4Birds dataset under various noise level (SNR, Signal-to-Noise Ratio) in Fig. IV). We can clearly see from this figure that DyDecNet exhibits better robustness to noise than the three comparing methods. Its dyadic decomposition strategy (differentiating between *approximation* and *detail* explicitly) and the proposed energy normalization help reduce noise interference.

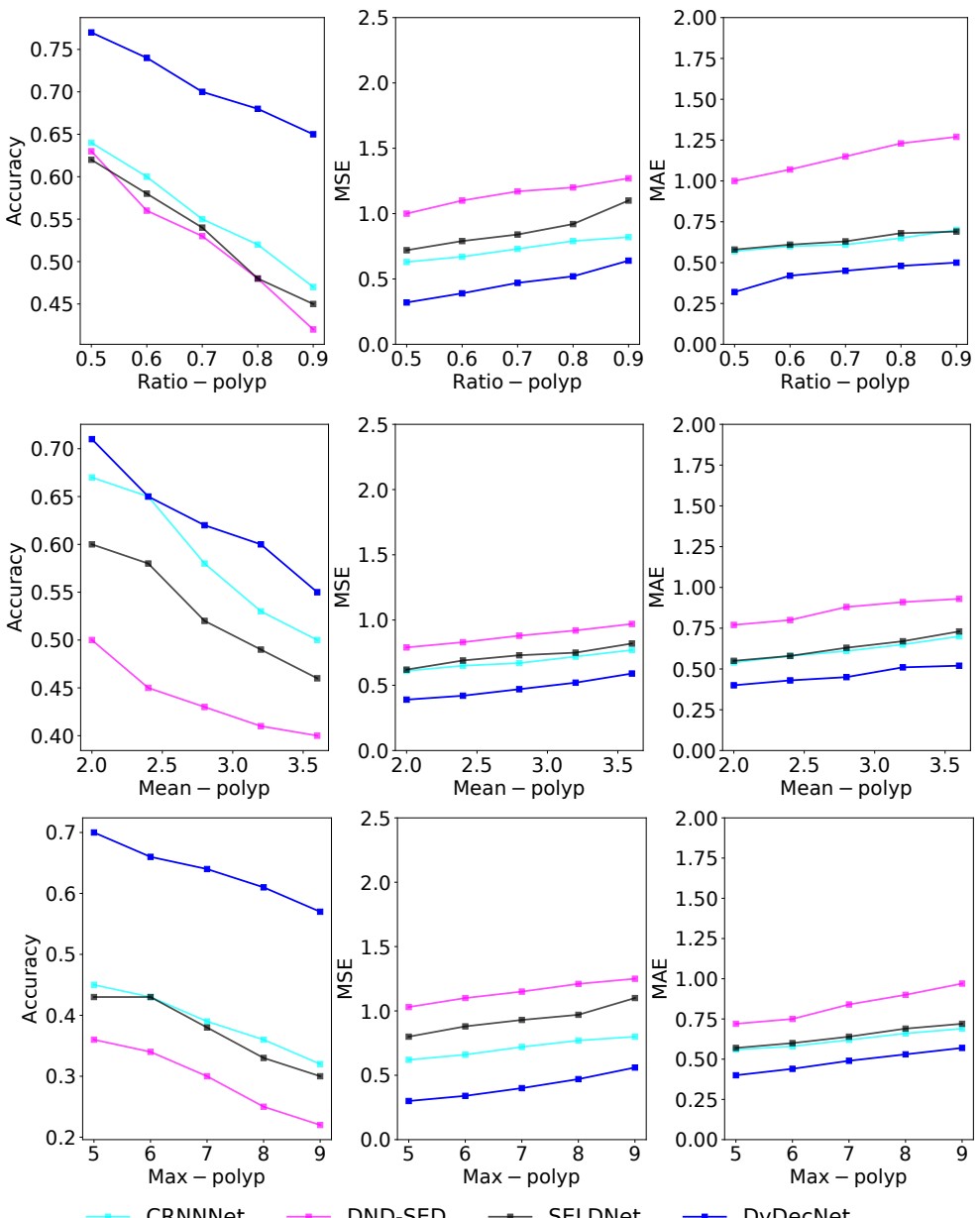

Figure III: AccuRate/MSE/MAE variation against $max$-polyp, $ratio$-polyp and $mean$-polyp on Polyphony4Birds Dataset.

## D.5 LEARNABLE ENERGY NORMALIZATION WITH TRADITIONAL T-F FEATURE

To test the efficiency of our proposed energy normalization module, especially combining them with traditional one-stage T-F features, we explicitly add one learnable energy normalization module just after the T-F feature extracted by traditional time frequency feature extractors. The comparison is given in Table VI and Table VII, from which we can observe that performance of traditional T-F feature slightly increases after introducing the energy normalization module. It thus shows the necessity of energy normalization for sound counting task in high-polyphonic situation. However, they still lead to inferior performance than DyDecNet, which shows hierarchically dyadic decomposition with energy normalization is essential for sound counting task.

Table IV: Dyadic Decomposition Neural Network Architecture Illustration. Input audio is 5 s long, sampling rate is 24 k Hz. The input waveform and intermediate features are in $[channelnum, height, width]$ format. In the dyadic decomposition front-end, the constructed filters' initialized trainable frequency cutoffs (high frequency cutoff and low frequency cutoff) evenly divide the frequency range of the input sound waveform (half of sampling rate). In our design, the filter number doubles as the "Depth" increases by 1. So each filter in the preceding depth associates with two child filters in the next depth, in which the two child filters carried frequency cutoffs evenly divide the frequency cutoff of their parent filter (see Fig. 1 in the main paper). In our implementation, we just connect a filter in the preceding depth with its two child filters in the next depth. We organize filters in channel dimension. In the dyadic decomposition front-end, each filter is instantiated as $Sinc(\cdot)$ filter, which comprises of a learnable high frequency cutoff and a low learnable frequency cutoff.

| Layer Name | Filter Num | Output Size |
|---|---|---|
| **Input Size**: 5 s audio waveform: [1, 1, $24000 \times 5$] | | |
| **Dyadic Decomposition Front-End** | | |
| Dyadic Decomp. Depth 1 | filter num = $2^1$, 2x downsampple | [2, 1, $12000 \times 5$] |
| Dyadic Decomp. Depth 2 | filter num = $2^2$, 2x downsampple | [4, 1, $6000 \times 5$] |
| Dyadic Decomp. Depth 3 | filter num = $2^3$, 2x downsampple | [8, 1, $3000 \times 5$] |
| Dyadic Decomp. Depth 4 | filter num = $2^4$, 2x downsampple | [16, 1, $1500 \times 5$] |
| Dyadic Decomp. Depth 5 | filter num = $2^5$, 2x downsampple | [32, 1, $750 \times 5$] |
| Dyadic Decomp. Depth 6 | filter num = $2^6$ | [64, 1, $750 \times 5$] |
| Dyadic Decomp. Depth 7 | filter num = $2^7$ | [128, 1, $750 \times 5$] |
| Dyadic Decomp. Depth 8 | filter num = $2^8$ | [256, 1, $750 \times 5$] |
| **Backbone Network** | | |
| Per-channel Pool | SincLowPass Filters, stride = 5 | [256, 1, $750 \times 1$] |
| Cross-channel Conv. | 1D Conv., filter num = 512 | [512, 1, $750 \times 1$] |
| Per-channel Pool | SincLowPass Filters, stride = 5 | [512, 1, $150 \times 1$] |
| Cross-channel Conv. | 1D Conv., filter num = 1024 | [1024, 1, $150 \times 1$] |
| Per-channel Pool | SincLowPass Filters, stride = 3 | [1024, 1, $50 \times 1$] |
| Cross-channel Conv. | 1D Conv., filter num = 512 | [512, 1, $50 \times 1$] |
| Cross-channel Conv. | 1D Conv., filter num = 256 | [256, 1, $50 \times 1$] |
| FC | FC, output_feat = 1 | [50, 1] |

Table V: MSE/MAE/AccuRate results on Multiple Bird Classes. The three values split by '/' in each entry indicate MSE, MAE and AccuRate, respectively. For the AccuRate, we just report the accuracy rate under $p = 0$. Following the experiment setting in the main paper, we run each model 10 times independently. We do not report the standard deviation, they are all within 0.003.

| Bird Classes | SELDNet [17] | CRNNNet [9] | DND-SED [16] | DyDecNet |
|---|---|---|---|---|
| 1 Bird | 0.89 / **1.19** / 0.33 | 0.87 / 1.16 / 0.35 | 1.04 / 1.27 / 0.24 | 0.54 / 0.85 / 0.55 |
| 4 Birds | 0.92 / 1.41 / 0.48 | 0.74 / 1.10 / 0.47 | 0.95 / 1.34 / 0.43 | 0.46 / 0.92 / 0.70 |
| 6 Birds | **0.88** / 1.40 / **0.49** | **0.72** / **1.09** / **0.49** | **0.93** / **1.26** / **0.48** | 0.43 / 0.90 / 0.74 |
| 8 Birds | 0.93 / 1.45 / 0.40 | 0.75 / 1.13 / 0.44 | 0.95 / 1.37 / 0.42 | **0.41** / 0.88 / **0.77** |
| 10 Birds | 0.95 / 1.49 / 0.37 | 0.78 / 1.17 / 0.39 | 0.97 / 1.40 / 0.37 | 0.44 / **0.82** / 0.72 |

# E   NETWORK ARCHITECTURE

The DyDecNet architecture is shown in Table IV.

# F   DENSITY MAP EXPLANATION

We provide more detailed illustration about density map in Fig. V. Specifically, the dyadic decomposition front-end and backbone network learn a 2D time-frequency feature representation (sub-figure A), in which the time dimension size is 50 (equals to 5 s audio length, each feature has a time resolution of 100 milliseconds). Such representation can be used to either regress the density map (as we

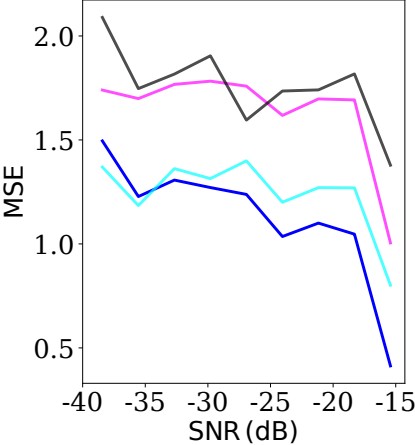

Figure IV: MAE variation under different signal-to-ratio. Different color indicates different methods, which aligns with the color in all other figures of the paper. Light Green: CRNNNet, Black: SELDNet, Magenta: DND-SED, Blue: DyDecNet. The horizontal axis goes (from left to right) with reduced noise interference.

Table VI: DyDecNet with Traditional T-F feature and learnable energy normalization module

| Method | MSE↓ | MAE↓ |
|---|---|---|
| DyDecNet_STFT | 1.30 | 1.46 |
| DyDecNet_LogMel | 1.27 | 1.47 |
| DyDecNet_MFCC | 1.26 | 1.44 |
| DyDecNet_Gabor | 1.30 | 1.43 |
| DyDecNet | **0.85** | **1.19** |

Table VII: DyDecNet with Traditional T-F feature

| Method | MSE↓ | MAE↓ |
|---|---|---|
| DyDecNet_STFT | 1.35 | 1.51 |
| DyDecNet_LogMel | 1.33 | 1.50 |
| DyDecNet_MFCC | 1.32 | 1.49 |
| DyDecNet_Gabor | 1.33 | 1.48 |
| DyDecNet | **0.85** | **1.19** |

do, sub-figure C) or classify the event class for each time frame (as sound event detection methods do, sub-figure E), by adding a full-connection layer (FC) to reduce the feature dimension size (can also be treated as frequency dimension) to 1. At the same time, we can use two full-connection layers to reduce the 2D representation to a scalar value, so as to directly regress the sound count number (sub-figure D).

The process of constructing density map is shown in sub-figure B. Please note that since we use supervised learning, we know each sound event start time and end time. So the density map can be easily constructed by setting the same value to the range between the start time and end time so that they are added up sound count number. Adopting density map for counting problem is widely used vision-based crowd counting tasks [49, 26], they show predicting density map usually give superior performance than object detection methods (in our case, SED method), and direct regressing count number. Their conclusion in vision-based crowd counting tasks matches our experimental result in sound-based counting tasks.

The reason why density map based method outperforms SED and direct regression lie two fold (according to our understanding): 1) unlike SED methods that try to discriminate different sound event class from temporally overlapping sound input, density map based method ignores sound event class but instead treat all sound events as an instance spanning their active time range. The reduced difficulty enables the neural network (DyDecNet) to learn expressive representations. 2) One characteristic of sound events is that each sound event has a certain start time and end time in the time dimension. The 2D time-frequency representation learned by the backbone network and dyadic decomposition front-end naturally maintain such characteristic. Using the 2D time-frequency representation to regress density map internally exploits the temporal location of various sound events, which helps sound counting task.

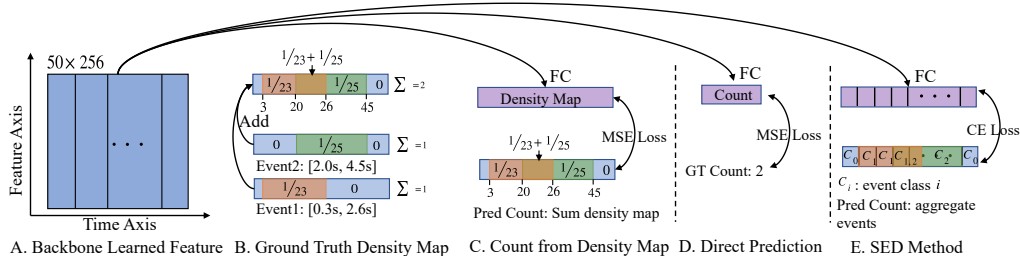

Figure V: Density map illustration. Each single sound event corresponds to an individual density map, which is a vector spans from the input audio length. The density value covers the whole event (connects its start time and end time) and sums to 1. For example, for the "Event1" with start time 0.3 s and end time 2.6 s in sub-figure B, its density map is a vector of 50, the values lie in [3, 26] are $\frac{1}{23}$. Finally, the overall density map for an input audio is obtained by summarizing all events' density map together. The sum (or integral) of the overall density map equals to the sound count number involved in the audio (in this case, 2). The comparison between three count methods are shown in sub-figure C, D and E, respectively. Given the feature representation learned by dyadic decomposition front-end and backbone network, we further use 1) a full-connection layers (FC) to reduce the channel dimension to 1 but keep the time dimension, so we can obtain a vector of the same density map size. We can either regress the density map (sub-figure C) or classify the event for each time step (SED method, it is multi-label classification, sub-figure E); 2) two full-connection layers (FC) to consecutively reduce both channel dimension and time dimension to 1, so that can directly predict the count number (sub-figure D).

