# OpenReview forum: "SoundCount: Sound Counting from Raw Audio with Dyadic Decomposition Neural Network"
_ICLR.cc/2023/Conference — Submitted to ICLR 2023_

### Official Review · Reviewer_6EcX · 2022-10-22

**Confidence:** 3
**Correctness:** 4
**Technical Novelty And Significance:** 3
**Empirical Novelty And Significance:** 3
**Recommendation:** 6

**Clarity, Quality, Novelty And Reproducibility:**

Clarity - Described Above. Presentation and flow could use work
Quality - Good experiments, ablation studies, results.
Reproducibility - Well described
Novelty - I don't know if the proposed learnable T-F filters has been done before. To my knowledge I haven't seen anything exactly the same, but I wouldn't be surprised if something similar exists in the literature.

**Strength And Weaknesses:**

The paper overall is good and interesting. The problem is one that has not been explored as thoroughly, and they are tackling highly polyphonic sounds in a way that hasn't been done before. The idea of learnable TF filters that grow exponentially with depth is interesting, and the ablation studies show that it is better than using other features like Mel or STFT features. The results seem pretty good, able to achieve solid counting accuracy with high numbers of concurrent sounds.

The clarity and organization of the paper could be improved significantly. For example, the section with Table 1-4 are very confusing. It should be possible to look at the tables and interpret them without searching for the description in the text. Table 1 is a dataset description, 2, and 3 are results, and 4 is an ablation study. But they are all presented as one unit, making it confusing. The caption for table 3 needs to say what dataset it was on, and table 4 should say "ablation study" and probably be included further down when the text is talking about the result. All the tables in general could do with a slightly more descriptive caption describe what result is being presented. I had to search through the text and back at the table to understand them.

I would also like to have some supplementary results to listen and compare  qualitatively . Hearing some of these datasets would be great (both for reviewers and listeners). It would also help us understand how difficult this problem is for human listeners. The ideal demo would be a video of highly polyphonic sounds with the prediction from your network being overlaid as the video plays.

**Summary Of The Paper:**

The authors propose DyDecNet, a new network architecture for counting sounds in highly polyphonic environments. The method contains a series of T-F band-pass filters which double in each layer to provide a learnable alternative to other frequency representation methods. It then contains a backbone and regresses the final count of sound sources. They also propose 3 new metrics for measuring polyphonic sounds.

**Summary Of The Review:**

Interesting approach, good results and experiments. Paper needs to be more clearly presented and qualitative results/examples would strengthen the work a lot by giving the necessary context for the problem.

---

### Official Review · Reviewer_5ud3 · 2022-10-23

**Confidence:** 2
**Clarity, Quality, Novelty And Reproducibility:** I think this work is interesting to m…
**Correctness:** 3
**Technical Novelty And Significance:** 3
**Empirical Novelty And Significance:** 3
**Recommendation:** 6

**Details Of Ethics Concerns:**

I think this work will lead to some privacy issue, since such technolgy is improperly used. It may lead to some negative social impacts.

**Strength And Weaknesses:**

I am not a person in acoustic data. Everything in this area is interesting to me. My comments for this work are listed as follows:
1. Can the author better claify the correlation bettwen this work and blind signal speration? It seems that this work has some close correlation with that problem.
2. I am a person in image separation. I know that such kinds of problems are very difficult.Even with different priors or assumption, we still cannot infer some highly mixed image contents. I think it should also be a problem for the acoustic issue. How do they address this problem in your work?
3. Do you need to consider the domain adaption issues? Or do you need to rely on some specific priors to achieve this goal about sound count?


**Summary Of The Paper:**

This work introduce a soundcount method. Their method is about how to conunt the number of distinct sound event.

**Summary Of The Review:**

I spend about 1 hour to review this paper. I just feel that the authors needs to provide more details about how do they solve the domain generalization problems.

---

### Official Review · Reviewer_3j8s · 2022-10-24

**Confidence:** 4
**Correctness:** 3
**Technical Novelty And Significance:** 3
**Empirical Novelty And Significance:** 4
**Recommendation:** 3

**Clarity, Quality, Novelty And Reproducibility:**

Clarity: The paper is written clearly and I was able to understand most of the details well. My only qualm is that a lot of details are relegated to the appendices and I find myself jumping back and forth often.
Quality: The quality of work done is satisfactory apart from my one major complaint about the authors misunderstanding a particular dataset/
Novelty: The architecture and task presented are both novel and the original.
Reproducibility: A lot of the details are present in the appendix which would enable the public to build the dydecnet model architecture. The supplementary materials contain the code as well. Not clear if the authors will release code to reproduce their results.

**Strength And Weaknesses:**

Strengths:

- The front-end network is novel and involving a multi-stage filterbank design for audio representation learning. The authors make a strong case for this frontend compared to traditional approaches for fixed or learnable filters for audio.
- The results obtained are strong compared to the baseline approaches.
- The ablation studies account for most if not all moving parts of the proposed model DyDecNet.
- The authors evaluate on a combination of real and synthetic datasets. The synthetic dataset is constructed using non-naive approaches involving room acoustics simulation and stochasticity.

Weaknesses:

- The Dydecnet architecture is presented as a generally more useful frontend for processing audio in neural networks, however it is only evaluated using sound counting as the task. It is not clear how this architecture is specifically useful for the counting problem. This paper has the potential to be a really strong paper had the authors not focused only on sound counting but also evaluated for other audio-related tasks as well, such as SED, source separation, etc.
- I am very familiar with the OpenMIC dataset for musical instrument classification. The authors mention that it contains the number of instrument events present in each clip. This is not true. The OpenMIC dataset is a partially labeled dataset where each audio clip is labeled with whether a subset from 20 instruments is present or not. As an example, there may be a clip which is labeled with 'vocals', 'guitar' and 'bass' present, and 'flute' absent. This does not mean that the clip has only 3 counts of sound events. It could have any number of notes being played by each instrument. Additionally, this also does not mean that the remaining 16 instruments are absent. It could very much be the case that there are 'drums' and 'violin' playing in the clip. They are just not labeled during the crowdsourcing campaign. This makes me question the validity of any experiments involving this dataset since the regression target is neither the number of sound events bring played, nor the number of unique instruments present. Perhaps a different music dataset should have been using such as MusicNet (https://zenodo.org/record/5120004).

**Summary Of The Paper:**

The paper discusses a new network architecture for counting the number of sound events in an audio signal, primarily focusing on bird calls but also evaluating with other sound classes. The main contribution of the work are two-fold:
1. A novel frontend for raw audio feature learning called dyadic frequency decomposition. This involves progressively increasing number of band-pass filters focusing on extracted information from high frequency and low frequency content of the parent filter's outputs. Additionally, a learned energy normalization layer is applied at each layer for improved loudness invariance. This approach converts raw waveform to a frequency representation using a multi-stage approach as opposed to other approaches which use a single-stage filterbank.
2. The authors propose new polyphony-based difficulty metrics for tasks involving polyphonic audio. They rate the datasets they use for evaluation using these metrics. They find that the results are consistent with the difficulty of the datasets.
Comparison against baselines from sound-event detection literature show that the so-called DyDecNet outperforms these baselines on most metrics. These comparisons are carried out using various modalities of audio data such as bird sounds, everyday sounds like phone calls, car engine noise, as well as music.
Various ablation studies show the benefits of the different techniques used in the network.

**Summary Of The Review:**

My overall impression is that the authors identified an issue in existing learnable audio frontends and proposed a new network architecture to address those issues. However, they demonstrate its effectiveness using the sound-counting task which is not a very popular area of research and could be considered a novel task. The issue lies in the fact that it is not clear how the new architecture is specialized for counting. The paper would be stronger if the evaluations included additional tasks like audio classification, source separation, etc.
Additionally, while I commend the authors including a music dataset since music is often highly polyphonic, based on my knowledge it would seem that there is a mismatch between the dataset chosen and the task the network is trained for as I have described in the weakness section. Given these points I am choosing a rating of 3.

---

### Official Review · Reviewer_xW8d · 2022-10-24

**Confidence:** 3
**Correctness:** 4
**Technical Novelty And Significance:** 3
**Empirical Novelty And Significance:** 3
**Recommendation:** 5

**Clarity, Quality, Novelty And Reproducibility:**

The paper is generally well-written and somewhat easy to follow, but it has some issues:
 - My main concern is that the paper is written in way that assumes that the dyadic approach is better than any other CNN-based approach by making bold statements without any reference to support them. For example: "Such shallow and one-stage processing is not powerful enough to learn robust representation where large loudness variance and heavy spectrum overlap exist in sound count task under high-polyphonic condition" (Sec 3). Moreover, there is no literature review on dyadic networks applied to other tasks that could support those claims. So I think it the paper should be rewritten slightly and should state that it is a hypothesis that the proposed approach is better, which will be then confirmed by the results.
- In line with my previous comment, a short literature review on dyadic networks should be provided.
- The experiments section is also quite hard to follow. I think it would be better to have one table which summarises all the results from the four corpora instead of splitting between the main paper and the appendix. The new metrics are also quite confusing: while they are sound for the task, they seem to never be used during the evaluation (only in Table 1). Please clarify.
- Some better motivation should also be added for the proposed architecture: why splitting the signal in two part (high and low frequencies) is better than a CNN, which could learn that by itself? I like the architecture, a bit more motivation would benefit the paper.
- In the Introduction, the footnote 1 seems contradictory to the statement it's attached to: "sound counting problem under highly polyphonic, cluttered and concurrent situation" and "To make sound countable, we focus on short, consistent and acoustically separable sound in this work.". So is the audio cluttered and concurrent, or consistent and separable? Please remove the footnote and clarify.
- In Section 3.3 the paper seems to imply that it is the first one to use 1D convolution: "Unlike existing methods [7, 1] that treat the frontend learned TF representation as a 2D image that is convolved by 2D Conv. operations, we treat it as a temporal representation so that we just need 1D Conv. to process it". This is misleading, as 1D convolution are widely used in audio and speech. Please clarify.
- The 5th paragraph of the Introduction starting with "Alongside the network, we propose three polyphony ..." need to be rewritten as it's not finished.
- I never saw the term "backbone" used here, wouldn't "backend" be better?
- The visualization comparison with MFCC is not helpful: It says that " traditional TF features (in our case, MFCC) encode cluttered and mixed TF representation that is much less visually separable" but MFCCs were never designed to be visually separable, and I don't understand why it matter. Please clarify or remove the visaulisation entirely.
 - In the Experiment section, the authors should be more objective with their statements: "Our framework shows excellence". This is discussable, as excellence is not well defined. Writing some like "is better than the baseline" is more objective.
- Some earlier references are missing for the raw speech literature review, like [1] and [2].

A part from those concerns, The proposed approach is novel as far as I can tell and the findings are significant as a new task is introduced, providing a good benchmark. The relevance of this task to the ICLR community is maybe limited.

[1] Palaz, D., Collobert, R., & Magimai-Doss, M. "Estimating Phoneme Class Conditional Probabilities from Raw Speech Signal Using Convolutional Neural Networks", proc. of Interspeech, 2013
[2] Tüske, Z., Golik, P., Schlüter, R., & Ney, H. "Acoustic modeling with deep neural networks using raw time signal for LVCSR", proc. of Interspeech, 2014

**Strength And Weaknesses:**

Strengths:
 - The paper introduces a new task, proposes a novel model to tackle it and provides thorough comparison with baseline models.
 - The paper introduces new metrics for the task.
 - The proposed approach is significant as it outperforms the baselines

Weakness:
 - The paper is not very clear:
      - It sometime makes some very bold and unsupported statements (see below)
      - Some references are missing and the proposed approach needs to be situate better with the existing works.
      - The experiments are a bit confusing.

**Summary Of The Paper:**

This paper introduces a new task, sound counting and a novel neural network architecture to tackle it based on a dual decomposition of the raw waveform. The proposed Dyadic model is first presented, along with new metrics for the task. The model is then evaluated on four partially synthezised datasets and compared with baseline methods. The proposed model is shown to outperforms the baselines on every dataset. Ablation studies are also presented, validating the different aspect of the architecture.

**Summary Of The Review:**

The paper introduces a new audio task and proposed a novel approach to solve it, which outperform the baselines. The architecture is novel as far as I can tell and the findings are significant, so I am keen to accept the paper if all of my concerns about clarity are addressed. But in the current form I recommend rejection.

---

### Official Review · Reviewer_6cJE · 2022-10-27

**Confidence:** 3
**Correctness:** 3
**Technical Novelty And Significance:** 3
**Empirical Novelty And Significance:** 3
**Recommendation:** 8

**Clarity, Quality, Novelty And Reproducibility:**

The paper describes many many concepts, and as a result, the writing is fairly dense in content, and somewhat difficult to keep track of.  However, the need for this density is understandable, given the potential queries that may arise during the review process.

There are many spelling and grammatical inaccuracies in the paper.

It may not be accurate to claim sound source counting as an underexplored problem, because of the existence of conferences with major tracks dedicated to acoustic event detection, sound source localisation, speaker diarisation, and speech separation, all of which involve sound source counting.  Thus, the existence of prior works related to the same problem in these tasks may somewhat limit the claimed novelty of this current paper.

The method proposed appears to be reproducible.  Several public datasets are used, which also allows for reproducibility.  However, the created synthetic datasets do not appear to be made publicly available, which therefore imposes difficulty of reproducing those specific experiments.


**Strength And Weaknesses:**

Strengths:
- The design of the proposed model and training, and the spectrum of datasets experimented upon, seem quite well thought through, to overcome many potential questions that may have otherwise arisen.  Specific justifying examples are:
-- The use of the hierarchical filtering binary tree seems to be well justified, based on an understanding of the expected acoustic properties of the sounds involved.
-- The separation of the model into feature extractor and backbone provides a natural taxonomisation of the model parts to different functions, while still being able to be jointly trained.
-- The model is able to count the sound sources both per-frame and per-session.
-- The use of energy gain normalisation shows an example of the authors carefully considering potential issues that may go wrong, for this specific task.
- The experiments are repeated on multiple diverse datasets, to demonstrate generalisable trends of the proposal.

Weaknesses:
- There is a lack of literature review to the highly related fields of speaker diarisation and sound source separation.  Sound source separation is only mentioned briefly in the conclusion, and no reference is given to related papers.  Speaker diarisation consists of two subtasks; 1) counting the number of speakers in an audio session, and 2) clustering the time segments of audio that belong to the same speaker together.  Sound source separation consists of two subtasks; 1) Counting the number of sources in an audio session, and 2) computing the audio of each separated source.  In both speaker diarisation and sound source separation, the two subtasks can either be done jointly within a single model or separately in a cascaded manner.  If done separately, then the first subtask is exactly the same task that this current paper is trying to address.  If done jointly, then the speaker count can be computed from the joint output, which indirectly also does the task that this current paper is trying to address.  Both speaker diarisation and sound source separation have several publications that consider neural network methods to either do these subtasks separately or jointly.  Thus, the existence of these works that have not been described in this current paper may somewhat reduce the novelty that this current paper claims.

- There are many spelling and grammatical inaccuracies in the paper.  Please proofread the paper.

- The created synthetic datasets do not seem to be made publicly available for experimental reproduction.  However, the paper also uses existing public dataset, which should allow the results to be experimentally reproducible.


**Summary Of The Paper:**

This paper proposes:
1) A novel model architecture and training method for the task of sound source counting.
2) Several novel measures to assess the difficulty of the sound source counting task.

The model architecture comprises 1) a feature extractor that operates hierarchically as a binary tree of cascaded filters, 2) a mechanism to pool across filters, and 3) 3 different methods to compute the output count from the pooled features.  Counting can be done both per-frame and per-session.

3 measures are proposed to assess the difficulty of the sound source counting task, all related to the amount of polyphony present in the audio.

Multiple datasets are used to assess the performance of the proposal, but public release of the created synthetic datasets is not reported in the paper for experimental reproduction.

Questions:
- What aspect of the model's performance is the chosen depth expected to impact?  Does the depth affect the effective maximum number of sources that the model can count?  Does the depth affect the frequency resolution that the model can rely on?  Does the depth affect the high frequency limit that the model can use?  How does the exponential scaling of the number of filters with depth affect the computational cost of training and inference?
- What are the sizes of the train and test splits of each dataset?  Such information is important to allow a reader to judge the applicability of the measured results.
- No reference is given to justify the motivating assumption of "human’s capability in discriminating different sound events reduces
seriously when the number of temporal-overlapping sound event number increases".
- In the statement "The child filter carrying the higher half frequency response
encode the parent’s processed intermediate waveform’s detail while the other one carrying the lower
half frequency response instead encodes the approximation", why does the higher half encode detail and the lower half encode approximation?  What do you mean by "approximation" and "detail"?

Ithenticate similarity score is 1%, which is good.


**Summary Of The Review:**

The claimed novelty of the task and model are somewhat diminished by the existence of prior works in diarisation and sound source separation.  However, the proposed task difficulty measures still appear fairly novel.  Taken together, the novelty in this paper should suffice.

Sufficient details are given in the paper for adequate experimental reproduction.

The proposed approaches seem well justified.

There is a need to further proofread the paper, to minimise spelling and grammatical inaccuracies.

---

### Author Response · Authors · 2022-11-18
**Thank you for all constructive feedbacks and positive comments**

We sincerely appreciate all reviewers’ constructive feedback and positive comments:

1. Novel architecture for solving sound source counting and novel measurements proposed. Reviewer **#6cJE**.

2. Novel task and novel model, new metrics for the task. Reviewer **#xW8d**.

3. Novel front-end design, strong experiment performance and lots of ablation study. Reviewer **#3j8s**.

4. Novel crowd counting task, comparing with image based crowd counting. Reviewer **#5ud3**.

5. New proposed task and novel T-F band pass filter design. Reviewer **#6EcX**.

Based on the reviewers’ comments, we have revised the manuscript and modification is highlighted in blue text colour. In summary, we have made the following five main modifications:

1. Revised several claims, either revised it with more supportive evidence, or mitigated the claim by constraining it within our scope and compared methods. (as raised concern from reviewer **#xW8d**).

2. Added more references and discussion on Dyadic Neural Network. (as raised concern from reviewer **#xW8d** and **#6cJE**, **#5ud3** and **#3j8s**).

3. Removed the footnote and figure visualization. (as raised concern from reviewer **#xW8d**).

4. Carefully re-organised the experiment section, in order to present it more legible and clear and organise it into a more coherent way. (as raised concern from reviewer **#xW8d**).

5. Clarified the OpenMIC dataset usage and task difference from other datasets. (as raised concern from reviewer **#3j8s**)

Moreover, we updated the supplementary material to include several sound clips to count, for a qualitative evaluation. (as raised concern from Reviewer **#6EcX**).

---

### Decision · Program_Chairs · 2023-01-20

**Decision:**

Reject

**Justification For Why Not Higher Score:**

As explained in the review above, this falls below the bar for publication at ICLR.

**Justification For Why Not Lower Score:**

N?A

**Metareview: Summary, Strengths And Weaknesses:**

This paper presents a model to count the occurrences of sounds in raw waveform audio. The task is correctly introduced, but the approach lacks a bit of motivation (for desgin choices), and is not perfectly situated with respect to prior work. Some of the experimental evaluation could be improved (e.g. OpenMIC is a federated dataset). In its current form, the paper is probably better suited for a signal processing, music, or an acoustic modeling workshop or focused venue.

**Summary Of Ac-Reviewer Meeting:**

There was no champion and we discussed the flaws and the scope, the seriousness of the flaws and if they were fixed in the rebuttal.